# Postnatal mechanical loading drives adaptation of tissues primarily through modulation of the non-collagenous matrix

Danae E Zamboulis[1]*, Chavaunne T Thorpe[2], Yalda Ashraf Kharaz[1], Helen L Birch[3], Hazel RC Screen[4†], Peter D Clegg[1†]

[1]Institute of Ageing and Chronic Disease, Faculty of Health and Life Sciences, University of Liverpool, Liverpool, United Kingdom; [2]Comparative Biomedical Sciences, The Royal Veterinary College, Royal College Street, London, United Kingdom; [3]University College London, Department of Orthopaedics and Musculoskeletal Science, Stanmore Campus, Royal National Orthopaedic Hospital, Stanmore, United Kingdom; [4]Institute of Bioengineering, School of Engineering and Materials Science, Queen Mary University of London, London, United Kingdom

**Abstract** Mature connective tissues demonstrate highly specialised properties, remarkably adapted to meet their functional requirements. Tissue adaptation to environmental cues can occur throughout life and poor adaptation commonly results in injury. However, the temporal nature and drivers of functional adaptation remain undefined. Here, we explore functional adaptation and specialisation of mechanically loaded tissues using tendon; a simple aligned biological composite, in which the collagen (fascicle) and surrounding predominantly non-collagenous matrix (interfascicular matrix) can be interrogated independently. Using an equine model of late development, we report the first phase-specific analysis of biomechanical, structural, and compositional changes seen in functional adaptation, demonstrating adaptation occurs postnatally, following mechanical loading, and is almost exclusively localised to the non-collagenous interfascicular matrix. These novel data redefine adaptation in connective tissue, highlighting the fundamental importance of non-collagenous matrix and suggesting that regenerative medicine strategies should change focus from the fibrous to the non-collagenous matrix of tissue.

*For correspondence:
danaez@liverpool.ac.uk

†These authors contributed equally to this work

Competing interests: The authors declare that no competing interests exist.

## Introduction

Functional adaptation of load-bearing tissues such as tendon is crucial to ensure the tissue is specialised appropriately to meet functional needs. Adaptation to mechanical requirements is key in healthy development and homeostatic tissue maintenance, with poor tissue optimisation during maturation likely a key contributor to increased injury risk later in life. Dysregulated homeostasis and long-term under- or over-stimulation leads to maladaptation, changes in tissue integrity, and reduced mechanical competence and is implicated in the disease aetiology of load-bearing tissues (*Freedman et al., 2015*; *Gardner et al., 2008*). Understanding the developmental drivers of structural specialisation and their association with mechanobiology is thus of fundamental importance for healthy ageing and disease prevention in musculoskeletal tissues (*Choi et al., 2018*; *Thorpe et al., 2013*). Such knowledge will help identify future targets for therapeutic interventions, and thus address the current lack of effective musculoskeletal disease treatments with new, evidence-based approaches to disease management. However, there is currently little knowledge of the key extracellular matrix (ECM) components associated with structural specialisation, the temporal nature of their adaptation, or the stimuli that drive adaptation.

As the principal structural component of connective tissues, collagen expression at the gene and protein level has been the focus of the majority of studies in relation to loading, with some studies reporting increases in collagen synthesis and others noting collagen degradation in response to loading, depending on the tissue function or tissue structure in different species (*Choi et al., 2018*; *Magnusson and Kjaer, 2019*). In tendons, this collagen structural framework is the fascicles and it is surrounded by the primarily non-collagenous and glycoprotein-rich components of the ECM, termed the interfascicular matrix (IFM) (*Figure 1—figure supplement 1b*; *Armiento et al., 2018*; *Thorpe and Screen, 2016*). This distinction is important, as it describes a fibre composite material, in which 'fascicle' and 'IFM' phases have different mechanical properties, and overall tissue mechanical properties and function are governed by the interplay of these two phases. When looking to understand functional adaption of a tissue, it is necessary to look at adaption of all ECM components.

Whilst fascicle and collagen adaptation has received some attention, adaptation of the IFM phase to mechanical stimuli remains poorly defined. Indeed, it is notable that the numerous studies investigating the mechanoresponsive nature of load-bearing tissues tend to restrict their focus to specific fibre or matrix components with no spatial distinction, and also focus on a single element of either structural or mechanical adaptation, such that limited information is gained (*Cherdchutham et al., 2001a*; *Choi et al., 2019*; *Mendias et al., 2012*). In order to provide the necessary complete profile of adaptive behaviour, it is crucial that phase-specific, temporospatial adaptation in the context of both structure and function is defined.

Identifying the drivers of adaptation requires use of a model system in which the temporospatial nature of adaptation can be fully profiled. Tendon provides the ideal model for such a study. It is well-established that mature tendons can present structural and mechanical specialisms (*Thorpe et al., 2015b*; *Thorpe et al., 2016a*; *Thorpe et al., 2016c*) and be grouped into two clear functional groups; stiff positional tendons, such as the anterior tibialis tendon and the equine common digital extensor tendon (CDET), that simply connect muscle to bone to effectively position limbs, whilst elastic energy-storing tendons, such as the Achilles tendon and the equine superficial digital flexor tendon (SDFT), are further specialised to provide an energy storing function, increasing locomotor efficiency by stretching and storing energy which they return to the system on recoil (*McNeill Alexander, 2002*; *Batson et al., 2003*; *Thorpe et al., 2012*; *Thorpe and Screen, 2016*). Further, the simple aligned organisation of tendon means that fascicle and IFM phases are spatially distinct, enabling structural and mechanical characterisation of each phase independently (*Thorpe et al., 2012*; *Thorpe and Screen, 2016*). Finally, use of equine tendon provides access to an exceptional model of adaptation. The SDFT has been shown to be highly analogous to the human Achilles tendon in its capacity for energy storage, injury profile and extent of specialisation and the anatomically opposing CDET is an example of a positional tendon, functionally comparable to the human anterior tibialis tendon (*Figure 1—figure supplement 1a*; *Biewener, 1998*; *Patterson-Kane and Rich, 2014*). Availability of samples enabled us to explore the extensive adaptation processes associated with late stage development, contrasting paired positional and energy-storing equine tendons through pre- and post-natal development.

Using this model, we investigate the process and drivers of functional adaptation, when tendons transition from an absence of loading (foetal: mid to end (6 to 9 months) gestation, and 0 days: full-term foetuses, and foals that did not weight-bear); through to weight-bearing (0–1 month) and then to an increase in body weight and physical activity (3–6 months; and 1–2 years). We hypothesise that early in development during gestation, the fascicle and IFM of functionally distinct tendons have identical compositional profiles and mechanical properties, with tissue specialisation occurring as an adaptive response to the mechanical stimulus of load-bearing, predominantly in the IFM of the elastic energy-storing tendon.

## Results

### Mechanical adaptation is localised to the IFM

First, we determined how the mechanical properties of the fascicle and IFM develop in tendon, with a particular focus on the temporospatial nature of mechanical adaptation and functional specialisation. Individual fascicles were dissected while an isolated region of IFM was tested by shearing

fascicles apart (*Figure 1—figure supplement 1c*). Samples were subjected to preconditioning followed by a pull to failure (*Figure 1—figure supplement 1c*). The yield point of samples was identified, denoting the point at which the sample became irreversibly damaged and was unable to recover from the applied load, and the sample failure properties also recorded, highlighting the maximum stress and strain the sample could withstand.

A significant increase in fascicle yield and failure properties was evident when comparing embryonic fascicles to those acquired immediately at birth (*Figure 1h–i and e–g*, respectively). However, data indicate minimal distinction in fascicle mechanics between functionally distinct tendons (*Figure 1e–i*) and, significantly, no specialisation for energy storage in response to loading during postnatal development.

Contrasting with fascicle mechanics, the failure properties of the IFM continued to alter throughout development with failure properties increasing markedly from 6 months onwards (*Figure 2e–g*). We also identify the emergence of an extended region of low stiffness at the start of the loading curve (i.e. an extended toe region) specific to the SDFT IFM pull to failure curve (*Figure 2b*). This indicates less resistance to extension, and together with the concomitant increase in IFM yield force and extension at yield (*Figure 2h–i*), demonstrates development of an overall greater capacity for extension in the SDFT IFM behaviour. A summary of these findings is achieved by plotting the amount of IFM extension at different percentages of failure force (*Figure 2j–k*), highlighting how the IFM of the energy-storing SDFT became significantly less stiff than that of the positional CDET during the initial toe region of the loading curve as the tendon adapts.

The viscoelastic properties of the developing IFM also showed significant interactions between tendon type and development, with IFM viscoelasticity significantly decreasing with development specifically in the energy-storing SDFT (*Figure 2a,c–d*), resulting in specialisation towards a more energy efficient structure.

## Structural adaptation is localised to the IFM

Having described the mechanical adaptation of the IFM to meet functional demand, we next performed a histological and immunohistochemical comparison of developing energy-storing and positional tendons to determine how temporospatial structural adaptation may dictate this evolving mechanical behaviour.

The energy-storing SDFT and positional CDET appeared histologically similar in the foetus, in both instances showing surprisingly poor demarcation of the IFM, which only became structurally distinct after birth and the initiation of loading (*Figure 3a*). Fascicle development was generally consistent in both tendon types with cellularity and crimp showing a reduction with development, cells displaying more elongated nuclei, and collagen showing a more linear organisation (*Figure 3b*, *Figure 3—figure supplement 1*, scoring criteria *Supplementary file 1*). In contrast, the IFM demonstrated divergence between tendon types with only the SDFT IFM showing an increase in cellularity following tendon loading and a retention of IFM width throughout development (*Figure 3b*, *Figure 3—figure supplement 1*).

The abundance of major ECM proteins was also generally consistent across fascicle and IFM in the foetus, with divergence of protein composition between phases only evident with further development (*Figure 4*). Notably adaptation was driven by changes in non-collagenous ECM components specifically, levels of which reduced in the fascicles and increased dramatically in the IFM through postnatal development (*Figure 4*, *Figure 4—figure supplement 1*). Of particular note, we demonstrate PRG4 (commonly known as lubricin) and TNC were predominantly found in the IFM of tendons and showed sparse staining or were absent, respectively, from the fascicles. We also demonstrate that elastin is preferentially localised to the IFM with its abundance decreasing only in the CDET with development. Furthermore, we show histological and compositional changes manifest after birth and with the initiation of loading, but that histological and compositional adaptation to loading then occurs over a period of months, involving both upregulation and downregulation of different histologic variables and ECM constituents.

## Adaptation relies on evolution of IFM composition only

To explore these concepts in further detail and to scrutinise the capacity for ECM adaptation, proteomic methodologies were adopted. With the mechanical and histological data identifying that

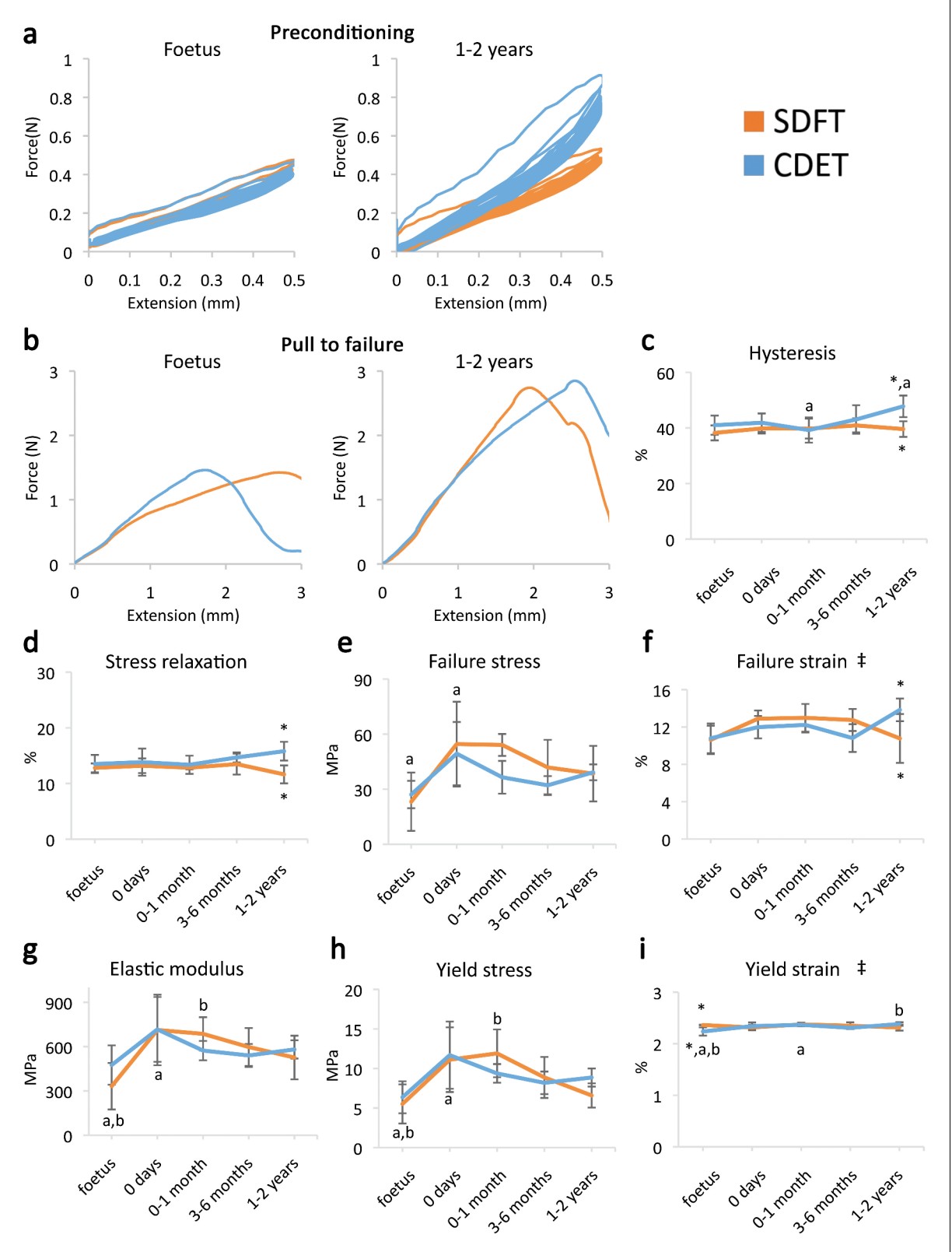

**Figure 1.** Fascicle response to mechanical testing shows increase in strength with development but few significant differences between tendon types, indicating that the fascicles show minimal structural specialisation in response to loading. (a) Representative curves for 10 preconditioning cycles for the SDFT and CDET fascicles in the foetus and 1–2 years age group. (b) Representative force-extension curves to failure for the SDFT and CDET fascicles in the same age groups. (c–i) Mean SDFT and CDET fascicle biomechanical properties are presented across development, with data grouped into age

*Figure 1 continued on next page*

*Figure 1 continued*

groups: foetus, 0 days (did not weight-bear), 0–1 month, 3–6 months, 1–2 years. ‡ significant interaction between tendon type and development, * significant difference between tendons, a-b significant difference between age groups. Error bars depict standard deviation. *Figure 1—figure supplement 1*. SDFT and CDET in the equine forelimb, tendon structure, and schematic showing procedure for biomechanical testing. The online version of this article includes the following source data and figure supplement(s) for figure 1:

**Source data 1.** Fascicle and IFM mechanical properties.
**Figure supplement 1.** SDFT and CDET in the equine forelimb, tendon structure, and schematic showing procedure for biomechanical testing.

functional adaptation is particular to the energy-storing SDFT, mass spectrometry analysis focused on a more detailed comparison of the IFM and fascicle development and adaptation in this tendon specifically.

Our results demonstrated that the proteomic profile of the IFM was more complex (more identified proteins) and a higher percentage of IFM proteins were cellular (*Figure 5—figure supplement 1*), supporting the histological findings of a more cellular IFM. Notably, despite the two phases being structurally distinct, they had 14 collagens and 11 proteoglycans in common (*Supplementary file 4*). Overall, proteomic heatmap analysis correlated very strongly with immuno-histochemical findings, showing that alterations in the fascicle proteome reduced through development, with minimal changes following the initiation of loading (*Tables 1* and *2*, *Figure 5a*), whilst numerous matrisome and matrisome-related proteins progressively increase in abundance through development in the IFM (*Tables 1* and *2*, *Figure 5a*). Detailed consideration of protein changes also highlights that post loading changes in the IFM appear more specific to proteoglycans and glycoproteins. Correlation analysis of IFM matrisome protein abundance and mechanical properties of the IFM across development revealed correlations for matrisome proteins abundance with the mechanical properties correlating IFM ECM composition and functional adaptation. Significant correlations included a negative correlation between proteoglycans DCN, LUM, OGN, PRELP and COL3A1 and the IFM hysteresis, a positive correlation between FBLN5 and stress relaxation, a positive correlation between COL6A1, COL6A2, and OGN and maximum stiffness, and for the yield properties a positive correlation of proteoglycans DCN, LUM, OGN, PRELP, and COL3A1 and force at yield point (*Supplementary file 5*). In addition, protein abundance for BGN, DCN, COMP, COL1A2, COL3A1 in the IFM and COMP and COL3A1 in the fascicles across development is mirrored by whole tendon mRNA expression (*Figure 5—figure supplement 2*).

Proteomic data also enabled insight into the turnover of proteins in the different tendon phases, through a comparison of the neopeptides produced by protein breakdown (*Thorpe et al., 2016b*). In the current study, we were able to profile the temporal nature of fascicle and IFM turnover, demonstrating that both phases display turnover during development but that fascicle turnover slows down towards the end of maturation, whilst IFM turnover rates are maintained, suggesting structural and/or compositional plasticity (*Figure 5b*).

Having identified the IFM as the location of functional adaptation, we next investigated the regulation of this process, to detect targets for modulation for regeneration strategies addressing functional impairment. For this purpose, pathway analysis was carried out for the differentially abundant proteins identified with mass spectrometry across age groups using the Ingenuity Pathway Analysis software (IPA). Pathway analysis revealed the canonical pathways 'integrin signalling' and 'actin cytoskeleton signalling' were predicted to be activated with development in the IFM supporting an ECM-integrin-cytoskeleton to nucleus signalling pathway for the mediation of the observed mitogenesis and matrigenesis in response to tendon loading. In addition, pathway analysis identified TGFB1 as an upstream regulator for the IFM dataset and based on the IFM protein abundance across age groups predicted TGFB1 to be inhibited in the foetus age group and to become activated in the 3–6 months age group. TGF-β1 was therefore highlighted as a potential regulator of ECM organisation and functional adaptation, predicted to be upregulated in the energy-storing tendon following loading (*Figure 6a*). This was supported by TGB1 mRNA expression in whole tendon increasing in the 3–6 months SDFT only, with the positional CDET TGFB1 expression showing no change with development (*Figure 6b*). In addition, knockdown of TGFB1 in equine adult tenocytes and stimulation with 10 ng recombinant TGF-β1 showed downregulation and upregulation, respectively, of key ECM components, BGN, COMP, COL1A2, and COL3A1, supporting a regulatory role for TGF-β1

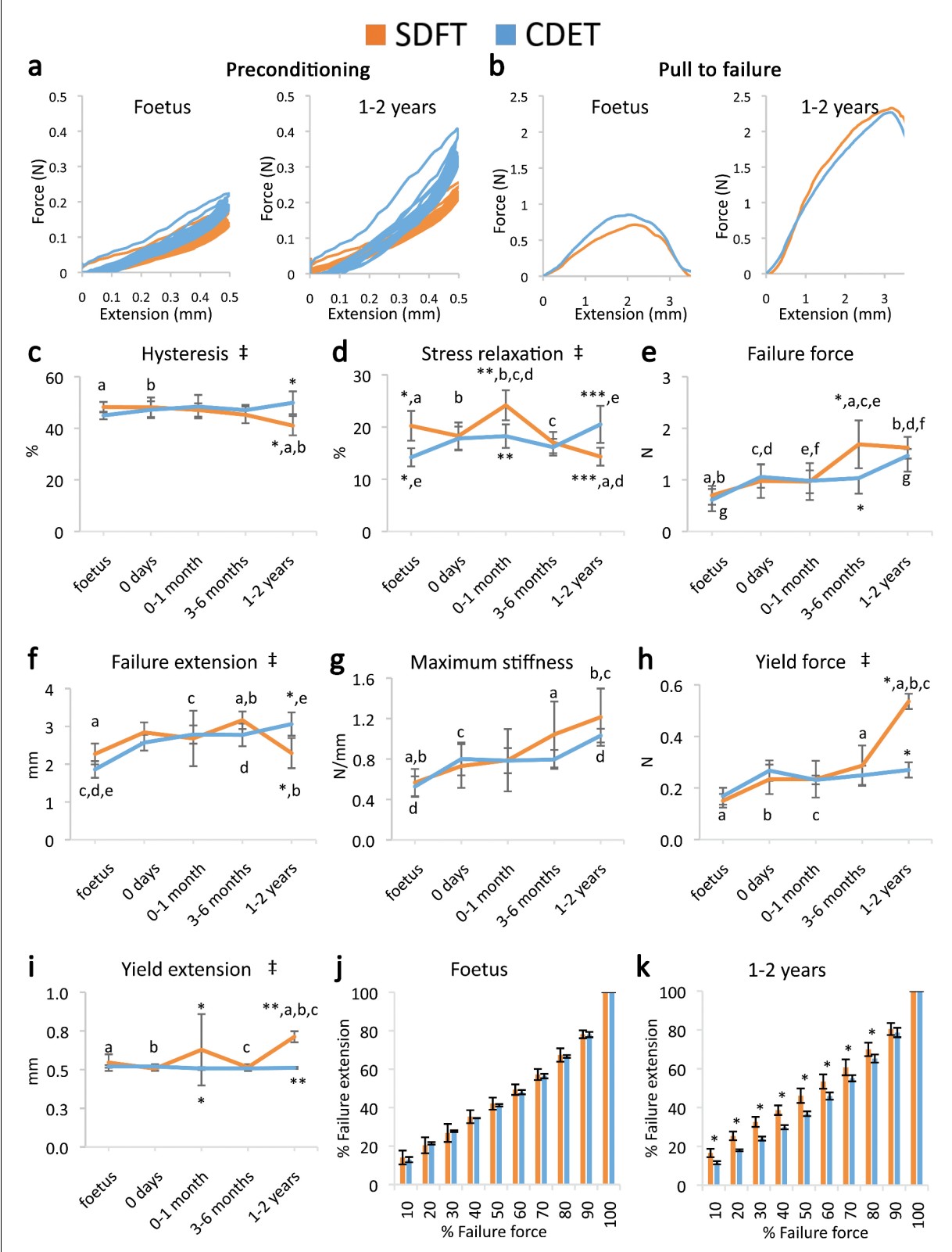

**Figure 2.** Mechanical testing of the IFM shows an equivalent increase in failure properties between the SDFT and CDET with development, but development of an extended low stiffness toe region and more elastic behaviour in the SDFT. (a) Representative curves for 10 preconditioning cycles for the SDFT and CDET IFM in the foetus and 1–2 years age group. (b) Representative force-extension curves to failure for the SDFT and CDET IFM in the same age groups. (c–i) Mean SDFT and CDET IFM biomechanical properties are presented across development, with data grouped into age

*Figure 2 continued on next page*

*Figure 2 continued*

groups: foetus, 0 days (did not weight-bear), 0–1 months, 3–6 months, 1–2 years. (**j–k**) To visualise the extended low stiffness toe region in the SDFT IFM, the amount of IFM extension at increasing percentages of failure force is presented, comparing the SDFT and CDET in the foetus and 1–2 years age group. ‡ significant interaction between tendon type and development, * significant difference between tendons, a-g significant difference between age groups. Error bars depict standard deviation.

(*Figure 6c–d*). Finally, correlation analysis of TGFB1 mRNA expression of whole tendon and IFM matrisome protein abundance across development revealed positive correlations with ECM proteins which were significant for COL1A2, COL2A1, COL4A1, COL4A2, COL6A3, HSPG2, and FN1 (*Supplementary file 6*).

## Discussion

In this study, we describe the phase-specific process and drivers of functional adaptation in tendon development integrating mechanical, structural, and compositional analysis in tendons transitioning

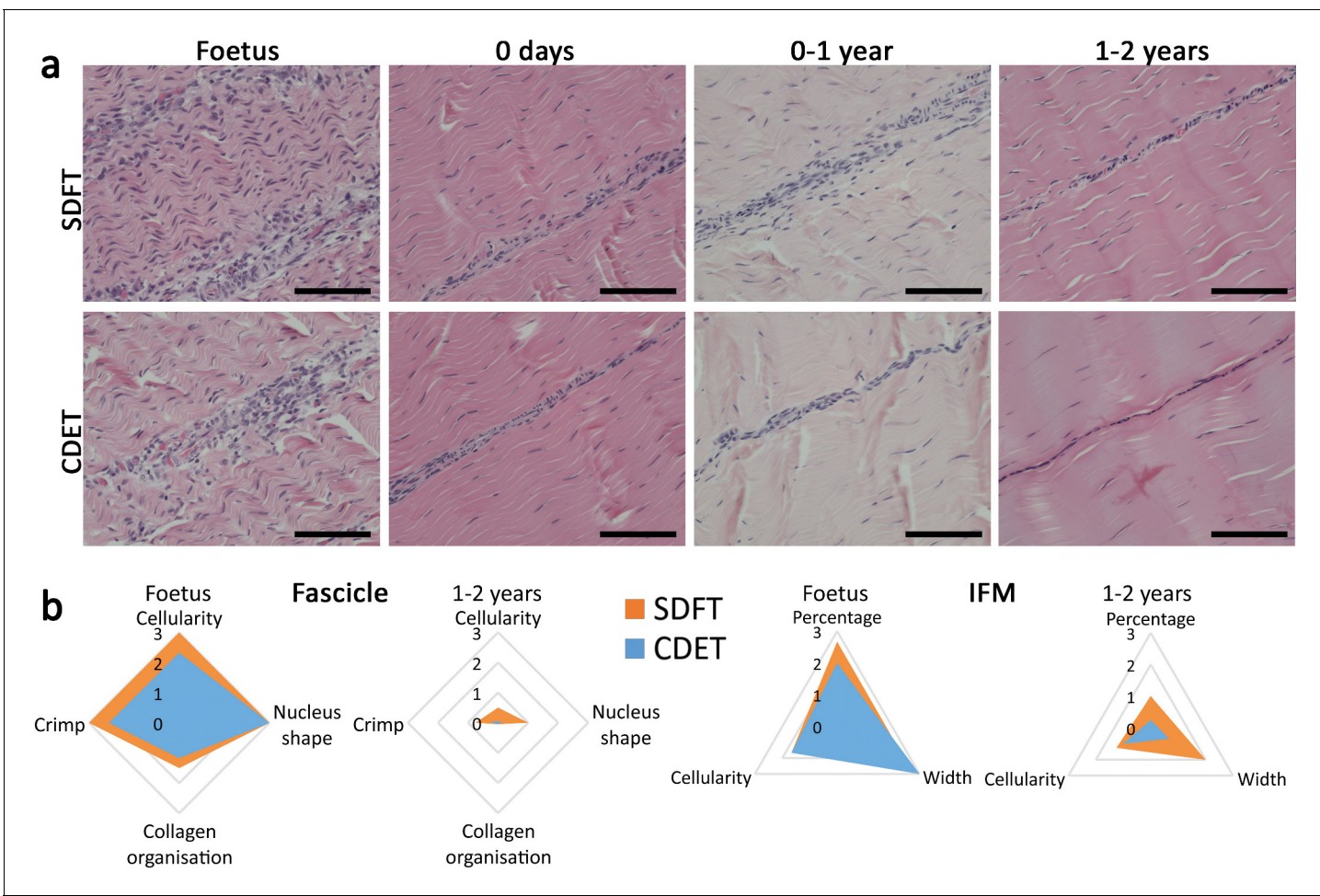

**Figure 3.** The SDFT and CDET are histologically similar at birth and differentiate with development especially in the IFM. (**a**) Representative images of H and E sections of the SDFT and CDET demonstrate structural development: foetus, 0 days (did not weight-bear), 0–1 year, and 1–2 years, whilst (**b**) Radar plots enable the mean histology scores of the fascicle and IFM for the SDFT and CDET to be compared between the foetus and 1–2 years age group (all data shown in *Figure 3—figure supplement 1* and scoring criteria in *Supplementary file 1*). A decrease in cell numbers, crimp, and IFM width is visible with progression of age, and the aspect ratio of cells in the fascicle increases. Scale bar 100 μm.

The online version of this article includes the following figure supplement(s) for figure 3:

**Figure supplement 1.** Scoring of histologic variables for the IFM and fascicle in the SDFT and CDET through postnatal development.

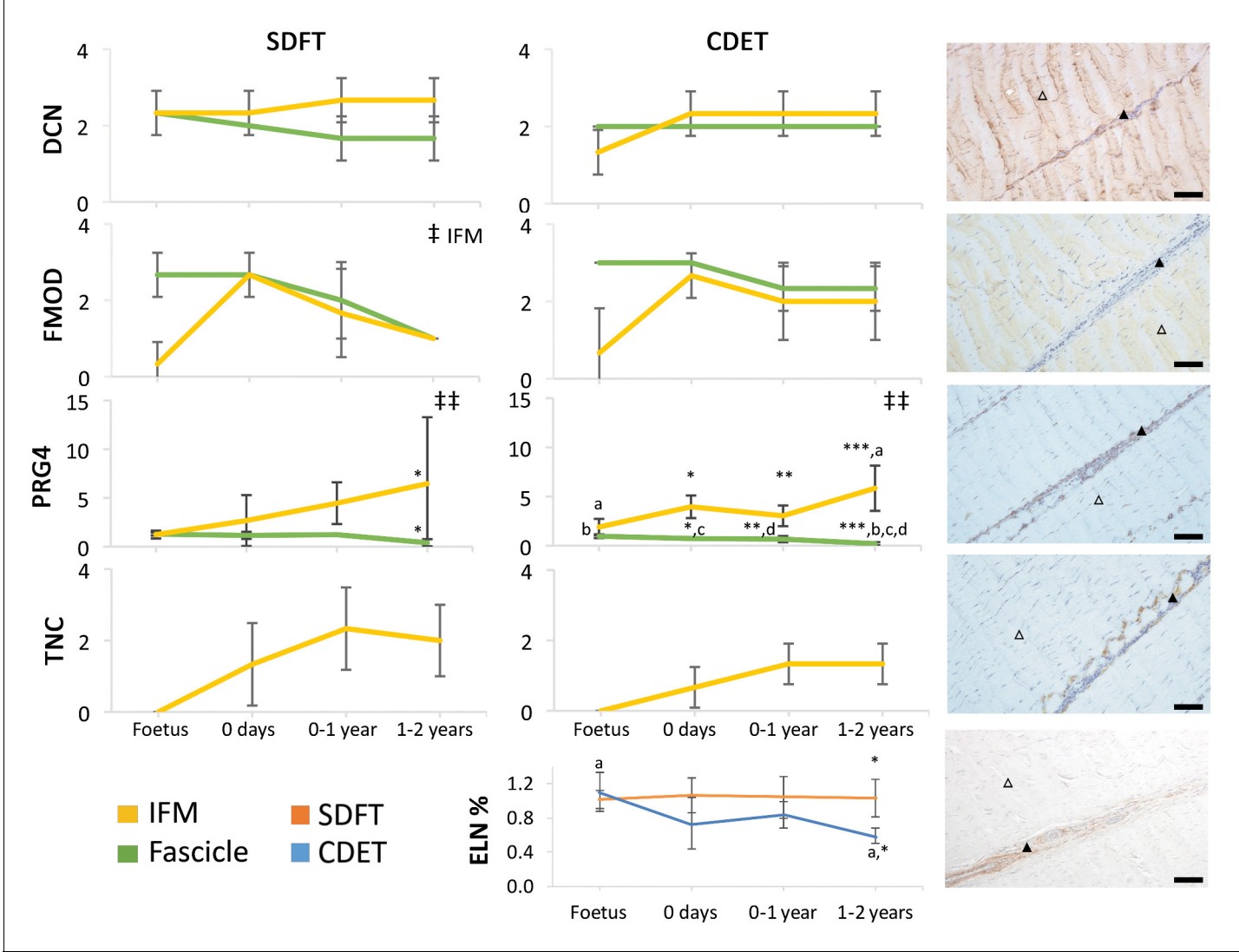

**Figure 4.** Immunohistochemical assays show divergence of PGR4 (lubricin) and elastin with maturation between functionally distinct tendons. IFM and fascicle staining scores are shown for decorin (DCN), fibromodulin (FMOD), lubricin (PRG4), and tenascin-C (TNC) in the SDFT and CDET, alongside representative images of immunohistochemical staining in the postnatal SDFT. DCN and FMOD staining is found in both IFM (black triangle) and fascicle (white triangle). PRG4 staining in mainly located in the IFM (black triangle) and less staining can be found in the fascicle (white triangle). TNC staining is restricted to the IFM (black triangle) and absent from the fascicle (white triangle). A quantitative measure of elastin (ELN) is provided as percentage of wet weight, alongside a representative image of immunohistochemical staining in the postnatal SDFT. ELN staining is mainly located in the IFM (black triangle) and faint staining can be found in the fascicle (white triangle). Staining scores for elastin are provided in *Figure 4—figure supplement 1*. ‡ significant change in tendon phase with development, ‡‡ significant interaction between tendon phase and development, * significant difference between tendons, a-d significant difference between age groups. Scale bar 100 μm. Error bars depict standard deviation. *Figure 4—figure supplement 1*. Scoring of ELN staining for the IFM and fascicle in the SDFT and CDET through postnatal development.

The online version of this article includes the following figure supplement(s) for figure 4:

**Figure supplement 1.** Scoring of ELN staining for the IFM and fascicle in the SDFT and CDET through postnatal development.

from an absence of loading through to weight-bearing and then to an increase in body weight and physical activity. To investigate functional adaptation and structure-function specialisation, we are contrasting fascicles and IFM of two tendons with distinct functions and mechanical properties; the equine SDFT and CDET. The energy-storing SDFT, which functions by stretching and recoiling with each stride to store and return energy, undergoes peaks strains recorded at 16.6% in vivo and has been found to be significantly more extensible than the CDET. The positional CDET, which functions to extend the distal limb prior to limb placement and is relatively inextensible to allow precise

**Table 1.** IFM differentially abundant matrisome and matrisome-associated proteins through development organised by highest mean condition (p<0.05, fold change ≥2).

Proteins are arranged into colour-coded divisions and categories. Bar graphs profile the relative abundance of each protein at each development stage, a foetus, b 0 days, c 0–1 month, d 3–6 months, e 1–2 years, with the development age reporting the highest mean protein level also specified.

| Protein | Division | Category | Highest mean cond. | a b c d e |
|---|---|---|---|---|
| SERPINH1 | Matrisome-associated | ECM Regulators | Foetus | |
| COL14A1 | Core matrisome | Collagens | 0–1 month | |
| ASPN | Core matrisome | Proteoglycans | 0–1 month | |
| FMOD | Core matrisome | Proteoglycans | 0–1 month | |
| KERA | Core matrisome | Proteoglycans | 0–1 month | |
| FBLN5 | Core matrisome | ECM Glycoproteins | 0–1 month | |
| FGB | Core matrisome | ECM Glycoproteins | 0–1 month | |
| FGG | Core matrisome | ECM Glycoproteins | 0–1 month | |
| COL1A2 | Core matrisome | Collagens | 3–6 months | |
| COL2A1 | Core matrisome | Collagens | 3–6 months | |
| COL4A1 | Core matrisome | Collagens | 3–6 months | |
| COL4A2 | Core matrisome | Collagens | 3–6 months | |
| COL6A3 | Core matrisome | Collagens | 3–6 months | |
| BGN | Core matrisome | Proteoglycans | 3–6 months | |
| HSPG2 | Core matrisome | Proteoglycans | 3–6 months | |
| ADIPOQ | Core matrisome | ECM Glycoproteins | 3–6 months | |
| FBN1 | Core matrisome | ECM Glycoproteins | 3–6 months | |
| FN1 | Core matrisome | ECM Glycoproteins | 3–6 months | |
| LAMB2 | Core matrisome | ECM Glycoproteins | 3–6 months | |
| LAMC1 | Core matrisome | ECM Glycoproteins | 3–6 months | |
| NID1 | Core matrisome | ECM Glycoproteins | 3–6 months | |
| ANXA4 | Matrisome-associated | ECM-affiliated | 3–6 months | |
| S100A4 | Matrisome-associated | Secreted Factors | 3–6 months | |
| COL21A1 | Core matrisome | Collagens | 1–2 years | |
| COL3A1 | Core matrisome | Collagens | 1–2 years | |
| COL5A1 | Core matrisome | Collagens | 1–2 years | |
| COL5A2 | Core matrisome | Collagens | 1–2 years | |
| COL6A1 | Core matrisome | Collagens | 1–2 years | |
| COL6A2 | Core matrisome | Collagens | 1–2 years | |
| DCN | Core matrisome | Proteoglycans | 1–2 years | |
| LUM | Core matrisome | Proteoglycans | 1–2 years | |
| OGN | Core matrisome | Proteoglycans | 1–2 years | |
| PRELP | Core matrisome | Proteoglycans | 1–2 years | |
| COMP | Core matrisome | ECM Glycoproteins | 1–2 years | |
| DPT | Core matrisome | ECM Glycoproteins | 1–2 years | |
| TGFBI | Core matrisome | ECM Glycoproteins | 1–2 years | |

The online version of this article includes the following source data for Table 1:
Source data 1. IFM and fascicle matrisome proteins intensity.

placement of the limb, experiences much lower strains than the SDFT (estimated at 2.5%) and is less extensible than the SDFT (*Batson et al., 2003*; *Birch et al., 2008*; *Thorpe et al., 2012*).

Whilst the limited previously available data on the development of tendon gross mechanical properties show an increase in mechanical properties with development (*Ansorge et al., 2011*; *Cherdchutham et al., 2001b*), no such phase-specific analysis of the development of tissue mechanics has been carried out previously. Similarly, available research into tendon morphogenesis and maturation has previously focused on the development of the collagenous network that comprises the tendon fascicles and is often focussed on early foetal development (*Kalson et al., 2011*; *Marturano et al., 2013*; *Pan et al., 2018*). Murine and zebrafish models used to investigate tendon development and adaptation have advanced our understanding of the control of fibrillogenesis by ECM proteins (*Subramanian et al., 2018*; *Subramanian and Schilling, 2014*; *Taye et al., 2020*), but these models lack an IFM, thus restricting our ability to explore the functional specialism we see in humans and other larger mammals.

Here, examining the fascicle and IFM mechanical properties independently, we show minimal distinction in fascicle mechanics between functionally distinct tendons and, significantly, no specialisation for energy storage in response to loading during postnatal development. In contrast, the IFM mechanical properties display continuous alterations through development with the properties of the IFM in the foetus being comparable between functionally distinct tendons, and a low stiffness region emerging in the initial non-linear region (toe region) of the pull to failure curve of the SDFT IFM only following tendon loading postnatally. This is coupled with a concomitant increase in IFM force and extension at yield, the point at which the sample became irreversibly damaged, highlighting that the energy-storing SDFT IFM becomes significantly less stiff than that of the positional

**Table 2.** Fascicle differentially abundant matrisome and matrisome-associated proteins through development organised by highest mean condition (p<0.05, fold change ≥2).

Proteins are arranged into colour-coded divisions and categories. Bar graphs on the right profile the relative abundance of each protein at each development stage, a foetus, b 0 days, c 0–1 month, d 3–6 months, e 1–2 years, with the development age reporting the highest mean protein level also specified.

| Protein | Division | Category | Highest mean cond. | A B C D e |
|---|---|---|---|---|
| COL11A1 | Core matrisome | Collagens | Foetus | |
| DCN | Core matrisome | Proteoglycans | Foetus | |
| FMOD | Core matrisome | Proteoglycans | Foetus | |
| KERA | Core matrisome | Proteoglycans | Foetus | |
| PCOLCE | Core matrisome | ECM Glycoproteins | Foetus | |
| SERPINF1 | Matrisome-associated | ECM Regulators | Foetus | |
| ANXA1 | Matrisome-associated | ECM-affiliated Proteins | Foetus | |
| ANXA2 | Matrisome-associated | ECM-affiliated Proteins | Foetus | |
| ANXA5 | Matrisome-associated | ECM-affiliated Proteins | Foetus | |
| LGALS1 | Matrisome-associated | ECM-affiliated Proteins | Foetus | |
| COL12A1 | Core matrisome | Collagens | 0 days | |
| COL3A1 | Core matrisome | Collagens | 1–2 years | |
| PRELP | Core matrisome | Proteoglycans | 1–2 years | |
| COMP | Core matrisome | ECM Glycoproteins | 1–2 years | |
| FN1 | Core matrisome | ECM Glycoproteins | 1–2 years | |

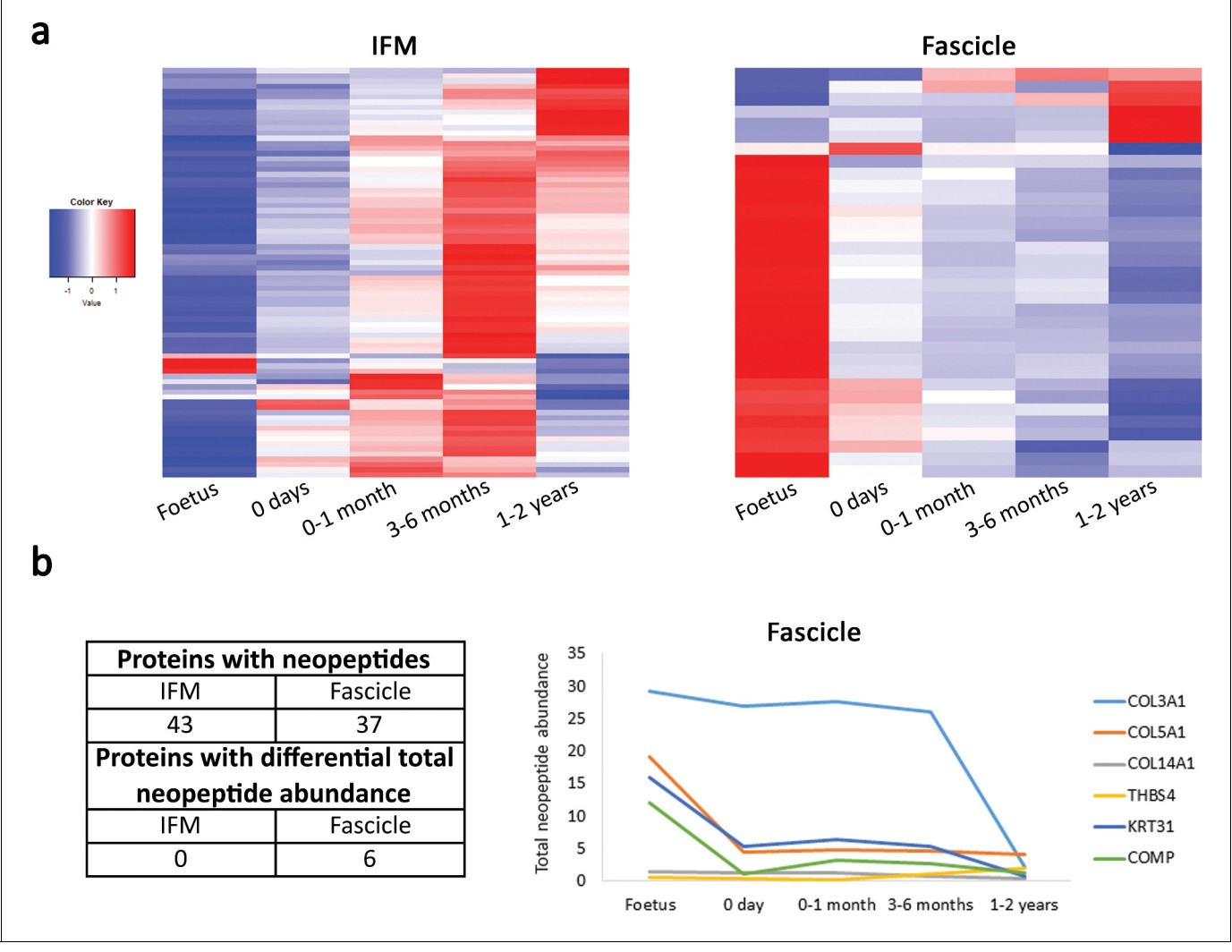

**Figure 5.** The fascicle proteome remains the same during postnatal development and tendon loading whereas the IFM proteome starts changing following tendon loading in postnatal development. (a) Heatmap of differentially abundant proteins in foetus, 0 days (did not weight-bear), 0–1 month, 3–6 months, and 1–2 years SDFT IFM and fascicles (p<0.05, fold change ≥2). Heatmap colour scale ranges from blue to white to red with blue representing lower abundance and red higher abundance. (b) Proteins with identified neopeptides and proteins showing differential total neopeptide abundance across age groups. Graph of proteins showing differential total neopeptide abundance in the SDFT fascicles across development (p<0.05, fold change ≥2, FDR 5%). No proteins showed differential total neopeptide abundance in the IFM. *Figure 5—figure supplement 1*. Classification of SDFT IFM and fascicle identified proteins and differentially abundant proteins according to their associated location. *Figure 5—figure supplement 2*. Relative mRNA expression of major ECM genes in whole tissue SDFT and CDET through postnatal development.
The online version of this article includes the following figure supplement(s) for figure 5:

**Figure supplement 1.** Classification of SDFT IFM and fascicle identified proteins and differentially abundant proteins (p<0.05, fold change ≥2) according to their associated location.
**Figure supplement 2.** Relative mRNA expression of major ECM genes in whole tissue SDFT and CDET through postnatal development.

CDET during the initial toe region of the loading curve. We have previously indicated that this low stiffness behaviour allows sliding between the fascicles, enabling non-uniform loading of tissue and is fundamental for effective extension and recoil in energy-storing tendons (*Thorpe et al., 2015b*). Furthermore, with ageing the low stiffness behaviour of the energy-storing IFM is lost, possibly contributing to disease development (*Thorpe et al., 2013*). The only other studies considering the mechanical properties of developing tendons have focused simply on changes in whole tissue mechanics, and have thus not been able to identify the drivers of change within the tissue (*Ansorge et al., 2011*; *Cherdchutham et al., 2001b*). Here, we identify that the IFM is the key

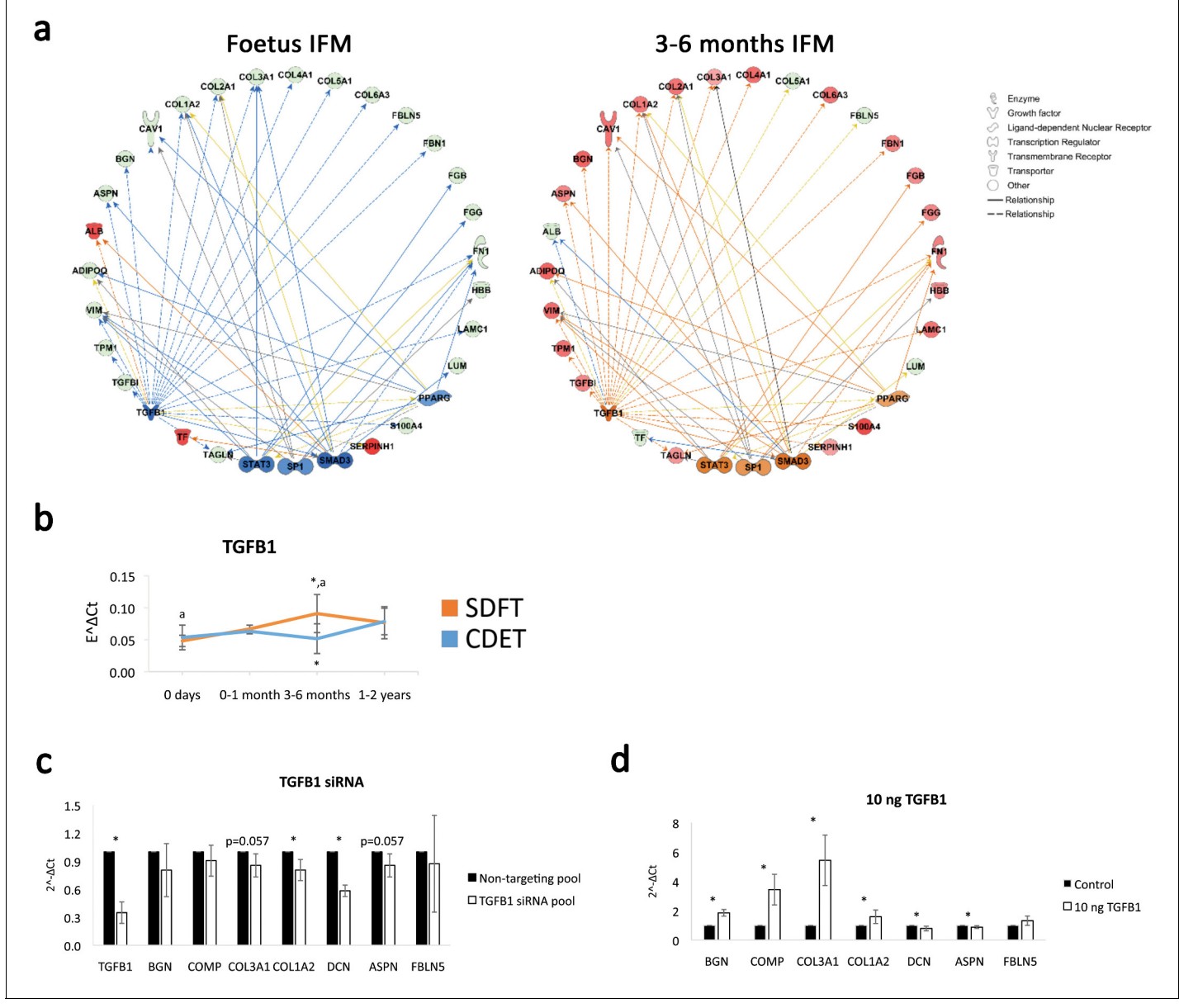

**Figure 6.** TGFB1 is predicted to be involved in compositional changes observed in the IFM. (**a**) IPA networks for TGFB1 as an upstream regulator were generated for the foetus and 3–6 months SDFT IFM proteomic datasets. TGFB1 regulation in the IFM is predicted to be inhibited in the foetus and activated at 3–6 months in the SDFT. Red nodes, upregulated proteins, green nodes, downregulated proteins, intensity of colour is related to higher fold-change, orange nodes, predicted upregulated proteins in the dataset, blue nodes, predicted downregulated proteins. (**b**) Whole tendon relative mRNA expression for TGFB1 in the SDFT and CDET during postnatal development shows an increase in TGFB1 mRNA in the 3–6 months highly loaded SDFT only. * Significant difference between tendons, a significant difference between age groups. (**c–d**) Relative mRNA expression of major ECM genes predicted to be regulated by TGFB1 in the IPA network following TGB1 knockdown (**c**) and stimulation with 10 ng recombinant TGF-β1 (**d**) for 24 hr. BGN, DCN, COMP, COL1A2 and COL3A1 show regulation following TGFB1 knockdown or addition. * Significant difference between control and treatment. Error bars depict standard deviation.

region in which mechanical adaptation to meet function occurs, and that this occurs after the initiation of loading, primarily 1–2 years postnatally.

We subsequently examine how temporospatial structural adaptation may dictate this evolving mechanical behaviour and uncover a divergence in structural characteristics between tendon types in the IFM only, with a retention of IFM width throughout development and an increase in cellularity in the SDFT IFM only. It is well recognised that foetal and early postnatal tendons are highly cellular,

and cellularity is generally considered to decrease postnatally (*Russo et al., 2015*; *Stanley et al., 2008*), but here we show that the described reduction in cellularity only occurs in the fascicles, and cellularity in fact increases in the IFM with development. By following the alterations in cellularity across IFM and fascicle, here we can determine that the marked difference in regional cellularity is likely driven by a maintenance of cell numbers following tendon loading in the thinning IFM, while cell numbers in the fascicle appear to reduce as a result of the fascicle ECM increase. Greater cellularity is commonly associated with a requirement for rapid adaptive organisation of ECM components (*Russo et al., 2015*), suggesting the IFM, particularly in the energy-storing tendons, may adapt to be more mechanoresponsive, a necessary aspect of healthy homeostatic maintenance of a tissue. Immunohistochemical analysis reveals the distribution of major ECM proteins is consistent across tendons in the foetus and becomes distinct across compartments through development. Of note, PRG4 (lubricin), a large proteoglycan which is important in ECM lubrication, is found mainly distributed in the IFM of tendons. Using a lubricin-knockout mouse, this proteoglycan has been demonstrated to facilitate interfascicular sliding (*Kohrs et al., 2011*), indicating that this structural adaptation may be key in achieving the previously identified mechanical adaptation in the energy-storing tendon IFM. In addition, *Kostrominova and Brooks, 2013* (*Kostrominova and Brooks, 2013*) report PRG4 expression as wells as elastin expression decreased with ageing in rat tendon suggesting an association with an increased risk of disease with ageing. We also demonstrate that elastin is preferentially localised to the IFM, potentially having a role in the capacity for matrix recoil after loading which is necessary for the healthy function of energy-storing tendons (*Godinho et al., 2017*; *Ritty et al., 2002*). Further supporting a role for elastin in the energy-storing function, elastin appears to be redundant in positional tendons with its abundance decreasing in the CDET with development. Together, these findings show that structural adaptation of tendon post loading is primarily focused to the IFM, and observed predominantly in the energy-storing SDFT.

Compositional analysis of the energy-storing SDFT compartments using mass spectrometry corroborates the immunohistochemical analysis and shows a more complex proteomic profile for the IFM. It additionally shows that abundance of the majority of proteins in the fascicles is higher in the foetus and reduces through development with minimal changes following the initiation of loading, whereas in the IFM numerous ECM and ECM-related proteins progressively increase in abundance following tendon loading and through development. Neopeptide analysis demonstrated ECM protein turnover in the fascicles slows down towards the end of maturation, whilst ECM protein turnover rates in the IFM are maintained. Once a tendon is mature, little collagen turnover occurs (*Birch, 2007*; *Heinemeier et al., 2013*) and we have previously shown that the minimal turnover in mature tendon is focused to the IFM (*Choi et al., 2020*; *Thorpe et al., 2010*; *Thorpe et al., 2015a*; *Thorpe et al., 2016b*). The maintenance of turnover rates observed in the IFM here suggests structural and/or compositional plasticity of the IFM. Integrated, our data convincingly show a continual temporal change in the IFM proteome specifically, which would enable adaptation and specialisation to the load environment, and highlight the compositional plasticity of the IFM in responding to dynamic altered conditions such as those occurring during development and regeneration. It is critical that the difference in capacity for functional adaptation across IFM and fascicle identified here is considered if regenerative medicine and tissue engineering approaches are to be successful. Here, we demonstrate the temporal pattern of structure-function adaptation, with compositional changes occurring in the first months after loading, and leading to the mechanical specialisation we have previously observed in adult energy-storing tendon (*Birch, 2007*; *Thorpe et al., 2012*; *Thorpe et al., 2015b*). With the fascicles primarily responsible for the mechanical strength of a tissue, biomaterial and regenerative medicine studies have unsurprisingly placed considerable emphasis on this region to date (*Sensini et al., 2019*; *Watts et al., 2017*). Here, we not only highlight the importance of the IFM in modulating mechanical behaviour, but also demonstrate how the IFM must be targeted to support adaptation and optimum tissue quality.

Finally, pathway analysis of our proteomic data highlighted TGF-β1 as a regulator of ECM organisation and functional adaptation, predicted to be upregulated following loading in the energy-storing tendon. TGF-β is a known regulator of proteoglycans and collagens in tendon (*Potter et al., 2017*; *Robbins et al., 1997*), a role we also demonstrated here with regulation of major ECM proteins mRNA expression following TGFB1 knockdown and stimulation in equine tenocytes. Further, TGFB1 mRNA expression was upregulated in the highly loaded energy-storing tendon only, supporting the hypothesis that TGF-β1 regulation is specific to the energy-storing tendon and subsequently

indicating that it may specifically be associated with loading. Exploring the specificity of TGF-β1 regulation and loading is challenging. Muscle paralysis interventions can be used to demonstrate a causal effect between mechanical force and TGF-β regulation (*Subramanian et al., 2018*), however, such experiments cannot be conducted in horses and other large mammals.

Whilst we acknowledge that other extrinsic factors may drive the changes, our direct comparison of the highly loaded SDFT with the low load CDET enables us to identify that the divergence in mechanical properties, adaptation, and TGF-β regulation all occurs only in the tendon experiencing significant loading. In addition, TGF-β has been shown to have a role in cellular mechanobiology and connective tissue homeostasis, regulating ECM synthesis and remodelling in a force-dependent way following mechanical stimulation, to specify the quality of the ECM and help coordinate cytoskeletal tension (*Maeda et al., 2011*; *Subramanian et al., 2018*). A previous study of developing chick tendon detected TGF-β1 staining in the IFM only, during development, highlighting its localised distribution in development (*Kuo et al., 2008*). In the current study, we are able to associate TGF-β expression also with functional adaptation of the tendon IFM. In addition to tissue development and homeostasis, TGF-β1 is involved in connective tissue injury and repair with abnormal expression levels reported in both processes suggesting a pleiotropic mode of action (*Gao et al., 2019*). The above may suggest a role for TGF-β1 in tissue development and homeostasis and that its dysregulation is associated with tissue injury and repair.

## Outlook

We demonstrate for the first time that functional adaptation in tendon is predominantly reliant on adaptation of the metabolically active IFM, which responds to the mechanical environment through TGF-β signalling, resulting in modulations in ECM turnover and composition to fine-tune mechanical properties. Traditionally, the non-collagenous matrix phase of connective tissues has received considerably less attention than the fibre phase, with regenerative medicine, biomimetics and biomechanics studies all largely focused on investigating and recapitulating the organisation and mechanical properties of the collagenous fibrous network.

Following tendon injury, normal tissue architecture is not recovered, and in particular, the cellular IFM is not regenerated. There is great potential gain from understanding the convergence of biology underpinning adaptation, function and pathology and here, we propose a paradigm shift to consider the metabolically active IFM as a key target for regenerative medicine strategies aimed at addressing functional impairment of tendons and other connective tissues following disease. Regeneration of the IFM following tendon injury could be key for tendon health and low re-injury risk.

## Materials and methods

### Key resources table

| Reagent type (species) or resource | Designation | Source or reference | Identifiers | Additional information |
|---|---|---|---|---|
| Biological sample (*Equus caballus*) | Superficial digital flexor tendon and common digital extensor tendon | Equine practices and commercial abattoir | | Foetus-2 years old |
| Biological sample (*Equus caballus*) | Primary superficial digital flexor tendon tenocytes | Commercial abattoir | | P3 from adult specimens |
| Antibody | Anti-decorin (mouse IgG) | Other | | (1:1500), Prof. Caterson, Cardiff University, UK |
| Antibody | Anti-proteoglycan 4 (mouse IgG) | Other | | (1:200), Prof. Caterson, Cardiff University, UK |
| Antibody | Anti-fibromodulin (rabbit IgG) | Other | | (1:400), Prof. Roughley, McGill University, Canada |
| Antibody | Anti-tenascin C (mouse IgG) | Santa Cruz Biotechnology | RRID:AB_785991 | (1:250) |
| Antibody | Anti-elastin (mouse IgG) | Abcam | RRID:AB_2099589 | (1:250) |

*Continued on next page*

*Continued*

| Reagent type (species) or resource | Designation | Source or reference | Identifiers | Additional information |
|---|---|---|---|---|
| Antibody | Zytochem Plus HRP polymer anti-mouse | Zytomed systems | RRID:AB_2868565 | (75 µL) |
| Antibody | Zytochem Plus HRP polymer anti-rabbit | Zytomed systems | RRID:AB_2868566 | (75 µL) |
| Sequenced-based reagent | *Equus caballus* TGFB1 Accell SMARTpool | Dharmacon, Horizon Discovery | https://horizondiscovery.com/en/products/tools/Custom-SMARTpool | (1 µM) |
| Sequenced-based reagent | *Equus caballus* Accell Non-targeting siRNA | Dharmacon, Horizon Discovery | https://horizondiscovery.com/en/products/tools/Custom-SMARTpool | (1 µM) |
| Peptide, recombinant protein | Recombinant Human TGF-β1 | Peprotech | 100–21 | (10 ng/mL) |
| Commercial assay or kit | FASTIN Elastin Assay | Biocolor | https://www.biocolor.co.uk/product/fastin-elastin-assay/ | |
| Chemical compound, drug | RapiGest SF | Waters | https://www.waters.com/waters/en_GB/RapiGest-SF-Surfactant/ | (0.1% w/v) |
| Software, algorithm | HistoQuest Analysis Software | Tissuegnostics | RRID:SCR_014823 | |
| Software, algorithm | Adobe Photoshop CS3 | Adobe | RRID:SCR_014199 | |
| Software, algorithm | Peaks Studio v8.5 | Bioinformatics Solutions | www.bioinfor.com/peaks-studio | |
| Software, algorithm | Ingenuity Pathway Analysis | Qiagen | RRID:SCR_008653 | |
| Software, algorithm | Matrisome | PMID:2197732 | http://matrisomeproject.mit.edu | |
| Software, algorithm | Mascot | Matrix Science | RRID:SCR_014322 | |
| Software, algorithm | Neopeptide Analyser | PMID:28503667 | https://github.com/PGB-LIV/neo-pep-tool/releases/ | |
| Software, algorithm | SigmaPlot | Systat Software Inc | RRID:SCR_003210 | |
| Software, algorithm | GProX | PMID:21602510 | RRID:SCR_000273 | |
| Other | Chondroitinase ABC from *Proteus vulgaris* | Merck | C2509 | (0.2 U/mL) |
| Other | Hyaluronidase from bovine testes | Merck | H3506 | (4800 U/mL) |

## Experimental design

Using an equine tendon model, we investigate the process and drivers of functional adaptation in the SDFT and CDET, two functionally distinct tendons, when tendons transition from an absence of loading (foetal: mid to end (6 to 9 months) gestation, and 0 days: full-term foetuses, and foals that did not weight-bear); through to weight-bearing (0–1 month) and then to an increase in body weight and physical activity (3–6 months; and 1–2 years). We use a phase-specific approach to characterise each tendon phase independently, by comparing fascicles (fibre phase) and interfascicular matrix (IFM; matrix phase) mechanical properties, structure and composition.

For this purpose, we used mechanical testing, histological and immunohistochemical analysis, and mass spectrometry analysis following laser capture microdissection. Sample size was selected based on previous experiments and restricted by sample availability and the cost of mass spectrometry analysis.

## Sample collection

Both forelimbs were collected from foetuses and foals aged 0–2 years (n = 19) euthanised for reasons unrelated to this project at a commercial abattoir or equine practices following owner consent

under ethical approval for use of the cadaveric material granted by the Veterinary Research Ethics Committee, School of Veterinary Science, University of Liverpool (VREC352). Collected tendons were split in the following age groups: Foetus (between 6 and 9 months of gestation; n = 4); 0 days (full-term foetuses (average gestation 11–12 months) and foals that did not weight-bear; n = 4): 0–1 month (n = 3); 3–6 months (n = 4); 1–2 years (n = 4).

The SDFT and CDET from one forelimb were dissected and wrapped in phosphate-buffered saline dampened tissue paper and foil and stored at −80°C for biomechanical testing. Two 1–2 cm segments from the mid-metacarpal area of the SDFT and CDET of the other forelimb were dissected, and one fixed in 4% paraformaldehyde for histology and immunohistochemistry, and the other snap frozen in isopentane and stored at −80°C for laser capture microdissection.

## Biomechanical testing of the fascicles

On the day of testing, samples were defrosted within their tissue paper wrap, then immediately prepared for testing. Fascicles were dissected from the mid-metacarpal region of the SDFT and CDET and subjected to a quasi-static test to failure according to *Thorpe et al., 2015a* (*Thorpe et al., 2015b*). Briefly, prior to testing, the diameter of each fascicle was measured along a 1 cm length in the middle of the fascicle with a non-contact laser micrometre (LSM-501, Mitotuyo, Japan, resolution = 0.5 µm) and the smallest diameter recorded and used to calculate cross-sectional area (CSA), assuming a circular shape.

Fascicles were loaded in an electrodynamic testing machine (Instron ElectroPuls 1000) equipped with a 250 N load cell and pneumatic grips (4 bar) coated with rubber and sandpaper to prevent sample slippage (*Thorpe et al., 2012*). The distance between the grips was set to 20 mm and fascicles preloaded to 0.1 N (approximately 2% fascicle failure load) to remove any slack in the sample. Following preload, the distance between the grips was recorded as the gauge length, then fascicles preconditioned with 10 loading cycles between 0% and 3% strain (approximately 25% failure strain) using a sine wave at 1 Hz frequency. Immediately after preconditioning, fascicles were pulled to failure at a strain rate of 5 %/s. Force and extension data were continuously recorded at 100 Hz during both preconditioning and the quasi-static test to failure. Acquired data were smoothed to reduce noise before calculations with a third-order Savitzky-Golay low pass filter, with a frame of 15 for the preconditioning data and 51 for the pull to failure data.

Using the preconditioning data, the total percentage hysteresis and stress relaxation were calculated, between the first and last preconditioning cycle. Failure force, extension, stress, and strain were calculated from the test to failure, and a continuous modulus calculated across every 10 data points of each stress strain curve, from which the maximum modulus value was determined. The point of maximum modulus was defined as the yield point from which yield stress and yield strain were determined.

## Biomechanical testing of the IFM

On the day of testing, tendons were defrosted within their tissue paper wrap, and IFM samples immediately dissected and prepared for biomechanical testing as described previously by *Thorpe et al., 2012*. Briefly, a group of two adjacent intact fascicles (bound by the IFM) were dissected, after which the opposing end of each fascicle was cut transversely 10 mm apart, to leave a consistent 10 mm length of intact IFM that could be tested in shear (*Figure 1—figure supplement 1c*).

Utilising the same electrodynamic testing machine and pneumatic grips as described for the fascicles, the intact end of each fascicle was gripped with a grip to grip distance of 20 mm, and a preload of 0.02 N (approximately 1% IFM failure load) applied. IFM samples were preconditioned with 10 cycles between 0 and 0.5 mm extension (approximately 25% failure extension) using a sine wave at 1 Hz frequency, then pulled to failure at a speed of 1 mm/s. Force and extension data were continuously recorded at 100 Hz during both preconditioning and the quasi-static test to failure. Acquired data was smoothed to reduce noise before calculations with a third-order Savitzky-Golay low pass filter, with a frame of 15 for the preconditioning data and 51 for the pull to failure data.

Total percentage hysteresis and stress relaxation were again calculated between the first and last preconditioning cycle. Failure force and extension were determined from the quasi-static pull to failure curve, and a continual stiffness curve was calculated across every 10 data points of the curve,

from which maximum stiffness was determined, and yield force and yield extension at maximum stiffness reported. Based on previous data demonstrating notable differences in the toe region of the IFM curve of functionally distinct tendons, the shape of failure curves was also compared between samples by calculating the amount of IFM extension at different percentages of IFM failure load (*Thorpe et al., 2015b*).

### Histology scoring

Paraformaldehyde-fixed paraffin-embedded longitudinal SDFT and CDET segments were sectioned at 6 μm thickness and stained with H and E for histologic examination and scoring (n = 13; three from each age group, four from 1 to 2 years age group). The examined histologic variables are reported in *Supplementary file 1* and adapted from *Nixon et al., 2008*.

For parameters scored by investigators, the sections were blinded and histologic variables assigned a grade from 0 to 3 by two independent investigators. Weighted Kappa showed moderate to good agreement in all instances, hence the average of the two scores was used. Other histologic variables were measured using image analysis (HistoQuest Analysis software, RRID:SCR_014823, Tissuegnostics and Adobe Photoshop CS3, RRID:SCR_014199) and then assigned a grade from 0 to 3. Cumulative scores for the fascicle and IFM for each horse were obtained by summing the scores of the fascicle and IFM variables, respectively, excluding IFM percentage to ensure IFM dimensions were not over-weighted in final reporting.

### Immunohistochemistry

Immunohistochemical analysis for DCN, FMOD, PRG4, TNC and ELN was carried out on paraformaldehyde-fixed paraffin-embedded longitudinal SDFT and CDET sections (6 μm thickness) (n = 12; three from each age group) as previously described by *Zamboulis et al., 2013*. Antigen retrieval was carried out with 0.2 U/mL Chondroitinase ABC (C2905, Sigma, Merck, Darmstadt, Germany) at 37°C for 2 hr for DCN, FMOD, PRG4, and TNC or with 4800 U/mL hyaluronidase (H3506, Sigma, Merck, Darmstadt, Germany) at 37°C for 2 hr for ELN. Primary antibodies were used at a concentration of 1:1500 for DCN (mouse IgG), 1:400 for FMOD (rabbit IgG), 1:200 for PRG4 (mouse IgG), 1:250 for TNC (mouse IgG, RRID:AB_785991, Santa Cruz Biotechnology, Dallas, Texas), and 1:100 for ELN (mouse IgG, RRID:AB_2099589, Abcam, Cambridge, UK). Antibodies for DCN and PRG4 were a kind gift from Prof. Caterson, Cardiff University, UK, and the FMOD antibody was kindly provided by Prof. Roughley, McGill University, Canada. The secondary antibody incubation was performed with the Zytochem Plus HRP Polymer anti-rabbit for FMOD and anti-mouse for DCN, PRG4, TNC, and ELN (RRID:AB_2868566 and RRID:AB_2868565, Zytomed Systems, Berlin, Germany). Immunohistochemical staining was graded from 0 to 3 (low to high) on blinded sections, assessing stained area and staining intensity for DCN, FMOD, TNC, and ELN. For PRG4, where staining was confined to the pericellular area, staining intensity was measured using HistoQuest Analysis software (Tissuegnostics, RRID:SCR_014823).

### Quantification of tendon elastin

The elastin content of SDFT and CDET samples from each age group (n = 12; three from each age group) was quantified using the FASTIN Elastin Assay (Biocolor, Carrickfergus, UK) (*Godinho et al., 2017*). Briefly, SDFT and CDET tissue was powdered (~15 mg wet weight) and incubated with 750 μl of 0.25 M oxalic acid at 100°C for two 1 hr cycles to extract all soluble a-elastin from the tissue. Preliminary tests showed two extractions were sufficient to solubilise all a-elastin from developing SDFT and CDET. Following extraction, samples and standards were processed in duplicate according to the manufacturer's instructions and their absorbance determined spectrophotometrically at 513 nm (Spectrostar Nano microplate reader, BMG Labtech, Aylesbury, UK). A standard curve was used to calculate the samples' elastin concentration and elastin was expressed as a percentage of tendon wet weight.

### Laser-capture microdissection

Laser-capture microdissection was used to collect samples from the fascicles and IFM of SDFT samples from all age groups (n = 4 for each age group with the exception of the 0–1 month group where n = 3). For this purpose, 12 μm transverse cryosections were cut from the SDFT samples and

mounted on steel frame membrane slides (1.4 µm PET membrane, Leica Microsystems, Wetzlar, Germany). Frozen sections were dehydrated in 70% and 100% ice-cold ethanol, allowed to briefly dry, and regions of fascicle and IFM laser-captured on an LMD7000 laser microdissection microscope (Leica Microsystems, Wetzlar, Germany) and collected in LC/MS grade water (FisherScientific, Hampton, New Hampshire). Collected samples were immediately snap frozen and stored at −80°C for mass spectrometry analysis.

## Mass spectrometry analysis

Mass spectrometry analysis of laser-captured SDFT fascicle and IFM samples was carried out as previously described by *Thorpe et al., 2016b*. Samples were digested for mass spectrometry analysis with incubation in 0.1% (w/v) Rapigest (Waters, Herts, UK) for 30 min at room temperature followed by 60 min at 60°C and subsequent trypsin digestion. LC MS/MS was carried out at the University of Liverpool Centre for Proteome Research using an Ultimate 3000 Nano system (Dionex/Thermo Fisher Scientific, Waltham, Massachusetts) for peptide separation coupled online to a Q-Exactive Quadrupole-Orbitrap mass spectrometer (Thermo Scientific, Waltham, Massachusetts) for MS/MS acquisition. Initial ranging runs on short gradients were carried out to determine the sample volume to be loaded on the column and subsequently between 1 and 9 µL of sample was loaded onto the column on a 1 hr gradient with an inter-sample 30 min blank.

## Protein identification and label-free quantification

Fascicle and IFM proteins were identified using Peaks 8.5 PTM software (Bioinformatics Solutions, Waterloo, Canada), searching against the UniHorse database (http://www.uniprot.org/proteomes/). Search parameters used were: peptide mass tolerance 10 ppm, fragment mass tolerance 0.01 Da, fixed modification carbamidomethylation, variable modifications methionine oxidation and hydroxylation. Search results for peptide identification were filtered with a false discovery rate (FDR) of 1%, and for protein identification with a minimum of 2 unique peptides per protein, and a confidence score >20 (-10lgp >20). Label-free quantification was also carried out using Peaks 8.5 PTM software for the SDFT fascicle and IFM separately. Protein abundances were normalised for collected laser-capture area and volume loaded onto the mass spectrometry column and differentially abundant proteins between the age groups in the SDFT fascicle and IFM were identified using a fold change ≥2 and p<0.05 (PEAKS adjusted p values). The mass spectrometry proteomics data have been deposited to the ProteomeXchange Consortium via the PRIDE partner repository, with the dataset identifier PXD012169 and 10.6019/PXD012169. With the IFM showing changes both in its protein composition and mechanical properties during development and TGFB1 being linked to protein composition, differentially expressed matrisome proteins identified in the IFM were correlated to TGFB1 whole tendon mRNA expression and the IFM mechanical properties using the Pearson correlation coefficient (p<0.05).

## Gene ontology and network analysis

The dataset of identified proteins in the SDFT fascicles and IFM were classified for cell compartment association with the Ingenuity Pathway Analysis software (IPA, RRID:SCR_008653, Qiagen, Hilden, Germany) and for matrisome categories with The Matrisome Project database (*Hynes and Naba, 2012*). Protein pathway analysis for the differentially abundant proteins between age groups in the SDFT fascicle and IFM was carried out in IPA. Protein interactions maps were created in IPA allowing for experimental evidence and highly predicted functional links.

## Neopeptide identification

For neopeptide identification, mass spectrometry data was analysed using Mascot server (Matrix Science, RRID:SCR_014322) with the search parameters: enzyme semiTrypsin, peptide mass tolerance 10 ppm, fragment mass tolerance 0.01 Da, charge 2+ and 3+ ions, and missed cleavages 1. The included modifications were: fixed carbamidomethyl cysteine, variables oxidation of methionine, proline, and lysine, and the instrument type selected was electrospray ionisation-TRAP (ESI-TRAP). The Mascot-derived ion score was used to determine true matches (p<0.05), where *p* was the probability that an observed match was a random event. The peptide list was exported and processed with the Neopeptide Analyser, a software tool for the discovery of neopeptides in proteomic data

(*Peffers et al., 2017*). Obtained neopeptide abundances for each sample were normalised for total peptide abundance for that protein and sample, and normalised neopeptide abundances were subsequently summed for each protein and the total neopeptide abundance analysed for differential abundance across the age groups in the SDFT fascicles and IFM using p<0.05 and FDR 5% (ANOVA and Benjamini-Hochberg FDR).

### Relative mRNA expression

Laser capture microdissection collects very small amounts of tissue which is not adequate for mRNA expression analysis and therefore whole tendon was used for the mRNA expression analysis. RNA extraction from whole SDFT and CDET was carried out followed by reverse transcription. Quantitative real-time PCR (qRT-PCR) was performed on an ABI7300 system (Thermo Fisher Scientific Waltham, Massachusetts) using the Takyon ROX SYBR 2X MasterMix (Eurogentec, Liege, Belgium). qRT-PCR was undertaken using previously validated gene-specific primers for DCN, FMOD, BGN, COMP, COL1A1, COL1A2, COL3A1, TGFB1, and GAPDH as a reference gene (*Peffers et al., 2013*; *Taylor et al., 2009*; *Supplementary file 2*). Relative expression levels were normalised to GAPDH expression and calculated with the formula $E^{-\Delta Ct}$ following primer efficiency calculation.

### SiRNA TGFB1 silencing and TGFb1 addition in tenocytes

Tenocytes isolated from young adult SDFT (passage 3, n = 4, average age: 5 years old) were transfected with custom Accell equine TGFB1 siRNA pool and an Accell non-targeting siRNA (Dharmacon, Horizon Discovery Ltd, Cambridge, UK) for 4 days to silence TGFB1. Experiments were carried out in the following 24 hr once TGFB1 knockdown was satisfactory. For TGFB1 stimulation, 10 ng/mL recombinant human TGFB1 (Peprotech, Cranbury, USA) was added to equine tenocytes for 24 hr, whilst control cells were incubated in the same media without any additions. qRT-PCR was undertaken as described above, using previously validated gene-specific primers for TGFB1, BGN, COMP, DCN, ASPN, FBLN5, COL1A2, COL3A1, and RPS20 as a reference gene (*Peffers et al., 2013*; *Taylor et al., 2009*; *Supplementary file 2*).

### Statistical analysis

Statistical analysis was carried out in SigmaPlot (RRID:SCR_003210, Systat Software Inc, San Jose, California) unless otherwise stated. Details of the n numbers for each experiment and the statistical test used for the analysis of the data are listed in *Supplementary file 3*. Heatmaps were designed in GProX (RRID:SCR_000273) (*Rigbolt et al., 2011*). The Central Limit Theorem (CLT) was used to assume normality where n > 30 and where n < 30 normality was tested using the Shapiro-Wilks test. If data were found not to be normally distributed their log10 transformation or ANOVA on Ranks was used for statistical analysis but the original data was presented in graphs.

## Acknowledgements

This work was funded by the Horserace Betting Levy Board, PRJ/776. We acknowledge the equine practices that provided samples for this study.

## Additional information

### Funding

| Funder | Grant reference number | Author |
| --- | --- | --- |
| Horserace Betting Levy Board | PRJ/776 | Danae E Zamboulis<br>Helen L Birch<br>Hazel R C Screen<br>Peter D Clegg |

The funders had no role in study design, data collection and interpretation, or the decision to submit the work for publication.

## Author contributions
Danae E Zamboulis, Conceptualization, Data curation, Formal analysis, Investigation, Visualization, Methodology, Writing - original draft, Writing - review and editing; Chavaunne T Thorpe, Conceptualization, Supervision, Methodology, Writing - review and editing; Yalda Ashraf Kharaz, Investigation, Writing - review and editing; Helen L Birch, Conceptualization, Supervision, Funding acquisition, Writing - review and editing; Hazel RC Screen, Peter D Clegg, Conceptualization, Resources, Supervision, Funding acquisition, Methodology, Project administration, Writing - review and editing

## Author ORCIDs
Danae E Zamboulis (iD) https://orcid.org/0000-0003-2839-7620
Chavaunne T Thorpe (iD) http://orcid.org/0000-0001-7051-3504

## Ethics
Animal experimentation: Samples were collected from horses euthanised for reasons unrelated to this project at a commercial abattoir or equine practices following owner consent under ethical approval for use of the cadaveric material granted by the Veterinary Research Ethics Committee, School of Veterinary Science, University of Liverpool (VREC352).

## Decision letter and Author response
Decision letter https://doi.org/10.7554/eLife.58075.sa1
Author response https://doi.org/10.7554/eLife.58075.sa2

# Additional files

## Supplementary files
• Supplementary file 1. Histologic variables used in the H and E scoring of the SDFT and CDET sections and the analysis method and reporting criteria adopted.

• Supplementary file 2. Gene primer sequences used in relative mRNA expression analysis.

• Supplementary file 3. Samples used for analysis along with statistical test used for analysis.

• Supplementary file 4. Collagens and proteoglycans identified in SDFT IFM and fascicle.

• Supplementary file 5. Correlation analysis of IFM protein abundance and mechanical properties across development.

• Supplementary file 6. Correlation analysis of TGFB1 whole tendon mRNA expression and IFM protein abundance across development.

• Transparent reporting form

## Data availability
The mass spectrometry proteomics data have been deposited to the ProteomeXchange Consortium via the PRIDE partner repository, with the dataset identifier PXD012169.

The following dataset was generated:

| Author(s) | Year | Dataset title | Dataset URL | Database and Identifier |
| --- | --- | --- | --- | --- |
| Zamboulis DE, Thorpe CT, Ashraf KY, Birch HL, Screen HRC, Clegg PD | 2018 | Equine energy-storing SDFT tendon development, Proteomic analysis of IFM and fascicles through development | http://proteomecentral.proteomexchange.org/cgi/GetDataset?ID=PXD012169 | ProteomeXchange, PXD012169 |

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
