## [Decision Letter]

**Acceptance summary:**

This manuscript addresses a key question of what key players drive changes in extracellular matrix in response to mechanical loading demand in tendon during postnatal growth. The authors findings that TGF-β signaling is actively involved in the organization of extracellular matrix and functional adaptation during mechanical loading provides a foundation for future evaluation of the role of TGF-β signaling for disease prevention in mechanoresponsive tissues. Additional data on the role of TGF-β signaling in extracellular matrix adaptation and revisions in response to the reviewer comments are found to be appropriate.

**Decision letter after peer review:**

Thank you for submitting your article "Postnatal mechanical loading drives adaptation of tissues primarily through modulation of the non-collagenous matrix" for consideration by *eLife*. Your article has been reviewed by three peer reviewers, including Subburaman Mohan as the Reviewing Editor and Reviewer #1, and the evaluation has been overseen by Clifford Rosen as the Senior Editor.

The reviewers have discussed the reviews with one another and the Reviewing Editor has drafted this decision to help you prepare a revised submission.

Summary:

While the reviewers have found the manuscript to be well written and the findings to be of interest to readers in the tendon biology field, the reviewers are in agreement that it needs to be revised by addressing the comments raised by the reviewers as below. In general, the reviewers found that while the mass spec data represents an advancement, it is not supported by functional analyses of the significance of the changes in TGF-β to tendon mechanical adaptation. The authors are invited to submit a revised version with additional experiments after a careful revision of the paper by addressing the comments given below. To guide the authors, it is critical for your group that 1) correlational analyses be performed between the expression and protein data from the mass spec data to support your major tenet, and 2) that knockdown of TGF-β in tenocytes be performed looking for changes in extracellular matrix proteins that would support the paper's major theme of functional effects of TGF-β. It is understood that other functional studies would be difficult in the equine model. Other concerns are noted below and can be addressed in a straightforward manner.

Reviewer #1:

In this study, Zamboulis et al. tested the hypothesis that the fiber and matrix phases of functionally distinct tendons (SDFT and CDET) have identical compositional profiles and mechanical properties during gestation and that tissue specialization in different tendons occurs as an adaptive response to mechanical loading. To test this prediction, they have studied the age-related changes in biochemical composition and structural adaptation using tendons from equine as model since they have been shown to be highly analogous to human Achilles tendons. The authors show that functional adaptation in tendons is predominantly reliant on adaptation of the metabolically active matrix phase and that TGF-β signaling is actively involved the ECM organization and functional adaptation following mechanical loading. This finding is interesting as one would have predicted a more predominant role for fibre phase since it is lot more abundant than the matrix phase.

Overall, the manuscript is well written, and the results are logically and clearly presented. While the authors highlight the importance of TGF-β as a regulator of ECM organization and functional adaptation based on its upregulation in the energy storing tendon following mechanical loading, data establishing the cause and effect relationship between changes in TGF-β, matrix composition and functional adaptation seem lacking and can be addressed in a number of ways including:

1) Performing correlation analyses between changes in TGF-β1 levels, matrix composition and functional changes in tendons during aging, and

2) Evaluating consequence of knockdown of expression or function of TGF-β1 in equine tenocytes on expression levels of key ECM proteins implicated in functional adaptation.

Reviewer #2:

The authors have conducted a well-designed study comparing the matricellular and fibrillar components of two tendons – SDFT and CDET at different developmental time points of development in horse. They have described in detail both the changes and differences in cellularity, mechanical properties, and composition of proteins and proteoglycans in both the phases of these tendons during development. From their descriptive study of tendons at these developmental time points, the authors propose that onset of mechanical force at birth has a role in bringing about significant changes in composition and cellularity of the matricellular phase of tendons. By using a mass spec followed by IPA, the authors conclude that TGF-β is the primary mechanical signaling pathway that is involved in bringing about these changes in tendon architecture and composition.

This study is of importance to the tendon community as it clearly shows the heterogeneity in fibrillar (collagenous) and matricellular (non-collagenous) phases of the tendon in composition, cellularity, and mechanical properties. The authors could improve the manuscript further by addressing the following concerns.

1) Previously published work in mice and zebrafish have shown that the matricellular protein- Thbs4 is essential for proper organization and bundling of collagen fibrils and levels of proteoglycans in the tendon (Frolova et al., 2014, Matrix Biology and Subramanian and Schilling, 2014) In addition to Maeda et al., 2011, a recently published work in zebrafish has shown that mechanical force controls cellular morphology and matrix organization in zebrafish tenons (Subramanian et al., 2018). The findings from these papers are relevant to the results shown by the authors and it would benefit the readers if the authors could discuss their results in the context of these published papers.

2) The authors state that based on their analyses, the changes in composition, functional adaptation, and biochemistry occur following mechanical loading in the matrix phase. The entire study relies on the analysis of tendons from developmental time points in horses. How are the authors ruling out other factors related to the normal development of a horse from birth? To be sure that mechanical force is involved, the muscles associated with the tendons should be paralyzed, which I believe is difficult to achieve in utero. Hence, the authors should modify their discussion accordingly.

3) Since *eLife* is a journal with wider readership, it would benefit the readers if there is a diagrammatic representation showing the respective tendons in the forelimb of the horse. The authors also have not defined the abbreviation CDET (subsection “Mechanical adaptation is localised to the matrix phase”, fourth paragraph).

4) The authors mention that based on pathway analysis an ECM-Integrin-Cytoskeleton to nucleus signaling paradigm was shown to be in action and they chose to focus on TGF-β signaling. The authors need to elaborate on how they arrived at TGF-β as the likely candidate considering that many ECM-Integrin interactions are possible in the tendon including several possible mechanotransduction pathways. Without a functional assay to support the hypothesis, the authors should explain if other components and pathways were considered and how they centered on TGF-β.

Reviewer #3:

Zamboulis et.al, using an unique model of equine tendon, investigated the process and drivers of functional adaptation, when tendons transition from an absence of loading in development during gestation through to weight-bearing after birth. The authors found that functional adaptation occurs postnatally, following mechanical loading, and is almost exclusively localised to the non-collagenous matrix phase, not fibre phase. The work provides important information on the understanding on the role of matrix phase in tendon adaptation. The experiments are generally well-designed. The authors should consider the following concerns:

1) The functional adaptation and tissue composition in superficial digital flexor tendon (SDFT) and common digital extensor tendon (CDET) were compared throughout the study. Figure 2 and Figure 3 show they have different adaptions after birth. More description/discussion should be included in the Introduction and Discussion sections on the main differences between these two types of tendon and why they show different adaptions.

2) The matrix phase of SDFT shows more significant changes in stress relaxation, failure force, failure extension than those of CDFT among all groups after birth. However, only PRG4 and ELN show remarkable differences in 1-2 years in SDFT and CDET, respectively. Explanation should be provided on the discrepancy.

3) The authors claimed that TNC is predominantly in the matrix phase of tendons. However, the data in Figure 3 does not support this conclusion. The peak score of TNC is around 2 in SDFT and CDET, which is comparable to scores of DCN and FMOD.

4) The histology data in Figure 3a and 3c are in poor quality. Higher resolution or higher power images are needed. In addition, the quantitation of the histology data are lacking in Figure 3a. It is not clear how the "more elongated nuclei" and "more linear organisation" were obtained. It would be helpful if the authors provide valid criteria for the measurements.

5) Detailed description /explanations are needed for results from Figure 4c. It would be helpful if more information are included in the figure legends or in the text.

6) Figure 3B and Figure 4C were not correctly cited in the text.

---

## [Author Response]

Reviewer #1:[…] Overall, the manuscript is well written, and the results are logically and clearly presented. While the authors highlight the importance of TGF-β as a regulator of ECM organization and functional adaptation based on its upregulation in the energy storing tendon following mechanical loading, data establishing the cause and effect relationship between changes in TGF-β, matrix composition and functional adaptation seem lacking and can be addressed in a number of ways including:1) Performing correlation analyses between changes in TGF-β1 levels, matrix composition and functional changes in tendons during aging, and2) Evaluating consequence of knockdown of expression or function of TGF-β1 in equine tenocytes on expression levels of key ECM proteins implicated in functional adaptation.

We are thanking the reviewer for the valuable comments and have carried out correlation analyses between TGFB1 mRNA expression, matrix composition and changes in the mechanical properties during development. The correlation analyses revealed some interesting correlations between the matrisome proteins and viscoelastic and yield point properties of the matrix phase (IFM). This work has been added in the Results section and the Supplementary files 5 and 6. We have also conducted TGFB1 knockdown and stimulation experiments to support the pathway analysis carried out with the Ingenuity Pathway analysis software. Knockdown of TGFB1 and stimulation with recombinant TGF-β1 showed downregulation and upregulation, respectively, of key ECM components, BGN, COMP, COL1A2, and COL3A1, supporting a regulatory role for TGF-β1 in tenocytes and strengthening the pathway analysis. This work has been added in the Results and Discussion section and in the Figure 6C-D.

Reviewer #2:[…] This study is of importance to the tendon community as it clearly shows the heterogeneity in fibrillar (collagenous) and matricellular (non-collagenous) phases of the tendon in composition, cellularity, and mechanical properties. The authors could improve the manuscript further by addressing the following concerns.1) Previously published work in mice and zebrafish have shown that the matricellular protein- Thbs4 is essential for proper organization and bundling of collagen fibrils and levels of proteoglycans in the tendon (Frolova et al., 2014, Matrix Biology and Subramanian and Schilling, 2014) In addition to Maeda et al., 2011, a recently published work in zebrafish has shown that mechanical force controls cellular morphology and matrix organization in zebrafish tenons (Subramanian et al., 2018). The findings from these papers are relevant to the results shown by the authors and it would benefit the readers if the authors could discuss their results in the context of these published papers.

Thank you for the addition of these excellent papers. In this study, we have been using the term “matrix phase” to refer to the interfascicular matrix interspersed around the fascicles of the tendon and the term “fibre phase” to refer to the tendon fascicles. We realise this could have created some confusion with “matrix phase” being used for all non-collagenous components of the tendon including the intrafascicular and interfibrillar matrix. To clarify this, we have replaced the term “matrix phase” with interfascicular matrix and the term “fibre phase” with fascicles throughout the manuscript.

Having been conducted in the zebrafish and the mouse the above papers are not directly related to the investigation of the interfascicular matrix as the interfascicular matrix is found in the tendons of humans and other larger mammals. However, although not directly investigating the interfascicular matrix phase to which we are referring in this study, these papers offer key additional information and support about the role of TGFB in the regulation of tendon organisation and adaptation as a response to mechanical force as well as the role of non-collagenous ECM components for the organisation of collagen in the fascicles and as such have been added to the Discussion.

2) The authors state that based on their analyses, the changes in composition, functional adaptation, and biochemistry occur following mechanical loading in the matrix phase. The entire study relies on the analysis of tendons from developmental time points in horses. How are the authors ruling out other factors related to the normal development of a horse from birth? To be sure that mechanical force is involved, the muscles associated with the tendons should be paralyzed, which I believe is difficult to achieve in utero. Hence, the authors should modify their discussion accordingly.

Muscle paralysis interventions can be used to demonstrate a causal effect with mechanical force; however, such experiments cannot be conducted in horses and other large mammals. Therefore, here, to help distinguish between the effect of normal development and changes related to loading during development we compared a highly loaded tendon, the superficial digital flexor tendon, and its positional counterpart, the common digital extensor tendon, which is not experiencing the same high loads. This way, divergence in adaptation and mechanical properties between the two tendons was attributed to events related to loading. The limitation of muscle paralysis experiments and how we overcame it was added to the Discussion. In addition, details about the difference in function and mechanical properties of the two tendons were added in the Introduction and Discussion to define the tendons better and make this clearer.

3) Since eLife is a journal with wider readership, it would benefit the readers if there is a diagrammatic representation showing the respective tendons in the forelimb of the horse. The authors also have not defined the abbreviation CDET (subsection “Mechanical adaptation is localised to the matrix phase”, fourth paragraph).

A definition for the CDET was added and details about the difference in function and mechanical properties of the two tendons were added in the Introduction and Discussion. In addition, we added a schematic of the equine forelimb with the SDFT and CDET highlighted in Figure 1—figure supplement 1A to aid understanding.

4) The authors mention that based on pathway analysis an ECM-Integrin-Cytoskeleton to nucleus signaling paradigm was shown to be in action and they chose to focus on TGF-β signaling. The authors need to elaborate on how they arrived at TGF-β as the likely candidate considering that many ECM-Integrin interactions are possible in the tendon including several possible mechanotransduction pathways. Without a functional assay to support the hypothesis, the authors should explain if other components and pathways were considered and how they centered on TGF-β.

The ECM-integrin-cytoskeleton to nucleus signalling is known to be involved in mediating cell responses to mechanical stimuli and integrin and actin cytoskeleton signalling were canonical pathways predicted to be activated in our dataset by pathway analysis (IPA). TGFB1 was not singled out from the ECM-integrin-cytoskeleton signalling but separately predicted to be an upstream regulator activated in our dataset by pathway analysis (IPA). The predicted canonical pathways and predicted upstream regulators were therefore two separate elements of the analysis rather than us choosing TGFB1 from the ECM-integrin-cytoskeleton signalling. Other components and pathways might be involved in tendon functional adaptation, however, based on our matrix phase proteomic dataset these pathways and regulators were highlighted in the analysis. This clarification was also added in the Results section.

Reviewer #3:[…] The authors should consider the following concerns:1) The functional adaptation and tissue composition in superficial digital flexor tendon (SDFT) and common digital extensor tendon (CDET) were compared throughout the study. Figure 2 and Figure 3 show they have different adaptions after birth. More description/discussion should be included in the Introduction and Discussion sections on the main differences between these two types of tendon and why they show different adaptions.

More details defining the function and the mechanical properties of the SDFT and CDET were added in the Introduction and Discussion and a schematic of the equine forelimb with the SDFT and CDET highlighted was added in Figure 1—figure supplement 1A to aid understanding.

2) The matrix phase of SDFT shows more significant changes in stress relaxation, failure force, failure extension than those of CDFT among all groups after birth. However, only PRG4 and ELN show remarkable differences in 1-2 years in SDFT and CDET, respectively. Explanation should be provided on the discrepancy.

Selected proteins were chosen for immunohistochemistry based on differences identified in mass spectrometry and on biologic importance. However, more proteins identified in our mass spectrometry dataset show differences between age groups that can account for changes in mechanical properties. In addition, more subtle differences may not be identified using immunohistochemistry where quantitation was based on scoring, with the exception of elastin which was quantified with a colourimetric assay and PRG4 which was quantified with the HistoQuest Analysis Software."

3) The authors claimed that TNC is predominantly in the matrix phase of tendons. However, the data in Figure 3 does not support this conclusion. The peak score of TNC is around 2 in SDFT and CDET, which is comparable to scores of DCN and FMOD.

Yes, TNC is predominantly found in the matrix phase/IFM and not found in the fibre phase/fascicles, rather than TNC is the predominant protein in the matrix phase/IFM. We modified this sentence in manuscript to make it clearer (subsection “Structural adaptation is localised to the IFM”).

4) The histology data in Figure 3A and C are in poor quality. Higher resolution or higher power images are needed. In addition, the quantitation of the histology data are lacking in Figure 3A. It is not clear how the "more elongated nuclei" and "more linear organisation" were obtained. It would be helpful if the authors provide valid criteria for the measurements.

The poor quality is due to the generation of the PDF and the actual figures have a higher resolution. Still, we split Figure 3 into two figures to allow for larger and clearer images for the previously labelled panels 3A and C. The full quantitation data are in Figure 3—figure supplement 1 along with the scoring criteria which are also found in Supplementary file 1. The method for obtaining the quantitation data is described in the Materials and methods (subsection “Histology scoring”).

5) Detailed description /explanations are needed for results from Figure 4C. It would be helpful if more information are included in the figure legends or in the text.

More information about the pathway analysis was added both in the legend of Figure 4C (now Figure 6A) and in the Results section.

6) Figure 3B and Figure 4C were not correctly cited in the text.

Amended manuscript citations for Figures 3B and 4C (now Figure 3B and Figure 6A).